# TRIM37-mediated stabilization of PEX5 via monoubiquitination attenuates oxidative stress and demyelination in multiple sclerosis insights from EAE and LPC-induced experimental models

Lai Jiang[1,2], Yue Jin[2], Yujia Han[2], Jin Fu[1]*

1 Department of Neurology, The Second Affiliated Hospital of Harbin Medical University, Harbin, China,
2 Department of Neurology, Beidahuang Group General Hospital, Harbin, China

* fujin@hrbmu.edu.cn

## Abstract

Multiple sclerosis (MS) is a chronic autoimmune disease of the central nervous system (CNS), characterized by myelin damage and neurodegeneration. This study focuses on the role of the TRIM37-PEX5 axis in regulating oxidative stress in oligodendrocytes and myelin repair, exploring its potential as a novel therapeutic target for MS. Through bioinformatics analysis, TRIM37 was found to be significantly downregulated in MS patients. *In vitro* experiments demonstrated that overexpression *TRIM37* could stabilize PEX5 protein via non-degradative monoubiquitination, thereby maintaining peroxisomal metabolic function, reducing oxidative stress levels, significantly decreasing apoptosis in both oligodendrocytes and neurons, and promoting the expression of myelin basic protein (MBP). Further mechanistic studies revealed that the TRIM37-PEX5 axis mitigates apoptosis in oligodendrocytes by regulating oxidative stress levels. *in vivo* experiments further confirmed the neurorestorative effects of TRIM37. In an experimental autoimmune encephalomyelitis (EAE) model, overexpression *TRIM37* significantly suppressed neuroinflammation mediated by microglia, reduced the expression of interleukin-1β (IL-1β) and tumor necrosis factor-α (TNF-α), and alleviated demyelination lesions (as evidenced by reduced myelin damage shown by Luxol fast blue (LFB) staining, $P<0.001$), while simultaneously increasing MBP expression levels ($P<0.001$). In conclusion, targeting the TRIM37-PEX5 axis holds promise as a novel strategy for improving myelin damage and providing neuroprotection in MS, offering a theoretical basis for interventions in metabolism-oxidative stress-related diseases.

## Introduction

Multiple sclerosis (MS) is a chronic autoimmune inflammatory disease of the central nervous system (CNS) and represents one of the most common causes of

**Data availability statement:** All relevant data are within the paper and its Supporting Information files.

**Funding:** Funding statement The research work presented in this article was supported by the 2024 Science and Technology Plan of the Health Commission of Heilongjiang Province. The project number is 20240303070234, with Jiang Lai serving as both the project leader and the recipient of the grant. The grant was awarded by the Health Commission of Heilongjiang Province.

**Competing interests:** The authors have declared that no competing interests exist.

**Abbreviations:** MS, Multiple sclerosis; CNS, Central nervous system; OPCs, Oligodendrocyte progenitor cells; WGCNA, weighted gene co-expression network analysis; GO, Gene Ontology; KEGG, Kyoto Encyclopedia of Genes and Genomes; PPI, protein-protein interaction; MBP, myelin basic protein; RA, retinoic acid; LPC, lysophosphatidylcholine; ROS, Reactive Oxygen Species; Co-IP, Co-Immunoprecipitation; CHX, cycloheximide; EAE, experimental autoimmune encephalomyelitis; CFA, complete Freund's adjuvant; PTX, pertussis toxin; DEG, differential expression gene.

non-traumatic disability in young adults aged 18–40 [1]. It is characterized by recurrent episodes and remissions of multifocal demyelinating lesions and neurological dysfunction [1–3]. The pathological core of MS involves the immune system attacking the myelin sheath in the CNS, leading to inflammation, demyelination, glial cell reactions, and axonal damage in both white and gray matter regions, ultimately resulting in reversible or irreversible neurological dysfunctions such as vision loss, limb weakness, and ataxia [2]. The prevalence of MS ranges from 5 to 300 cases per 100,000 individuals, with approximately 2.8 million people worldwide affected by the disease. The female-to-male ratio is 3:1, and the average age at diagnosis is 32 years. The chronic accumulation of physical and cognitive impairments in MS patients significantly impacts social, economic, and personal quality of life [1,4,5].

Mechanistically, MS involves the abnormal activation and imbalance of T cells, B cells, myeloid cells, and glial cells within the CNS. These cells contribute to CNS inflammation, oligodendrocyte demyelination, and neuronal injury by secreting pro-inflammatory cytokines such as IFN-γ, IL-17, and GM-CSF, as well as by disrupting the blood-brain barrier [6–10]. Demyelinating lesions are a characteristic pathological feature of progressive MS, and demyelination leads to impaired nerve conduction and axonal degeneration, which are the primary causes of disability in MS patients [11–13]. Myelin is synthesized by oligodendrocytes in the CNS and Schwann cells in the peripheral nervous system and wraps around neuronal axons in a spiral fashion, with interruptions at the nodes of Ranvier, enabling the rapid saltatory conduction of action potentials along the axon [13,14]. Remyelination not only facilitates the restoration of nerve conduction but also possesses neuroprotective effects, making it crucial for the treatment of MS and other demyelinating diseases [15,16]. Oligodendrocyte progenitor cells (OPCs) are key players in remyelination, but the inflammatory microenvironment in MS patients inhibits their proliferation, migration, and differentiation [8,12,13,17]. Although remyelination can partially repair demyelinating lesions, the regenerated myelin is often thin and incomplete, and its capacity declines with disease progression, age, and persistent inflammation, ultimately leading to repair failure and exacerbated neurodegeneration [14,18]. Current therapies for MS primarily focus on addressing CNS inflammation, neuroprotection, and ultimately achieving remyelination [19,20].

Mounting evidence indicates that oxidative stress and peroxisomal dysfunction are key drivers of demyelination and neurodegeneration in MS. Peroxisomes play a critical role in neutralizing reactive oxygen species (ROS) and maintaining lipid homeostasis [21,22], and their impairment has been linked to increased oxidative damage in oligodendrocytes and accelerated myelin loss in MS patients. However, the molecular mechanisms underlying peroxisomal dysfunction in MS remain poorly understood. The *TRIM37* gene is located on chromosome 17q23. TRIM37 contains a RING finger domain and functions as an E3 ubiquitin ligase. Recent studies have revealed its significant role in the development of various tumors [23–27]. Moreover, it has been reported that TRIM37 localizes to peroxisomes, and its mutations can lead to peroxisomal dysfunction [28]. PEX5 is the receptor for peroxisomal matrix proteins, responsible for importing proteins into peroxisomes by recognizing PTS1 and PTS2 signals.

TRIM37 can specifically recognize and add a monoubiquitin label to lysine K464 on PEX5, which is crucial for maintaining PEX5 stability and facilitating the import of peroxisomal matrix proteins [29]. However, the specific role of TRIM37 in MS pathogenesis—particularly its connection to oxidative stress, peroxisomal function, and demyelination—has not been previously explored. Given the central role of oxidative stress in MS and the link between TRIM37 and peroxisomal-mediated antioxidant defense, we hypothesized that TRIM37 may serve as a critical regulator of myelin stability in MS, making it a rationale target for investigation. Here, our study for the first time reveals that TRIM37 expression is significantly downregulated in MS patients. TRIM37 stabilizes the PEX5 protein through monoubiquitination, improves peroxisomal function, and reduces oxidative stress and apoptosis in oligodendrocytes, thereby ameliorating MS.

Therefore, this study aims to: (1) validate the functional role of TRIM37 downregulation in MS pathogenesis; (2) elucidate the mechanistic link between TRIM37-mediated PEX5 monoubiquitination and peroxisomal dysfunction in oligodendrocytes; (3) assess the therapeutic potential of targeting the TRIM37-PEX5 axis in alleviating oxidative stress, demyelination, and neuroinflammation; and (4) establish TRIM37 as a novel target for MS intervention through both in vitro and in vivo models.

## Materials and methods

### Bioinformatics analysis

To systematically identify key genes associated with the pathological progression of MS, we selected and analyzed the gene expression dataset GSE135511 from the NCBI-GEO database (https://www.ncbi.nlm.nih.gov/geo/). This dataset comprises brain white matter tissue samples from 40 MS patients and 10 healthy controls. Data preprocessing was performed using R language (v4.2.0) and the limma package. Through differentially expressed gene (DEG) analysis (adjusted P-value < 0.05, |log2FC| > 1), we identified genes significantly upregulated or downregulated in the MS group and visualized their distribution and significance using a volcano plot. To elucidate gene co-expression patterns, weighted gene co-expression network analysis (WGCNA) was employed to identify gene modules (blue module) significantly associated with MS lesions, from which core genes were extracted. Gene Ontology (GO) and Kyoto Encyclopedia of Genes and Genomes (KEGG) enrichment analyses were conducted on the DEGs and WGCNA key genes using the clusterProfiler package. To optimize disease prediction biomarkers, machine learning and random forest algorithms were utilized to rank key genes, which were further filtered using the Gini coefficient. A protein-protein interaction (PPI) network was constructed using the STRING database (confidence score > 0.7), providing a bioinformatics basis for subsequent experimental mechanistic studies.

### In *Vitro* model establishment

**Cell culture.** Human microglial cells (HMC3), human oligodendrocytes (MO3.13), and human neurons (SH-SY5Y) were purchased from the Cell Resource Center of the Shanghai Institute of Life Sciences, Chinese Academy of Sciences. HMC3 cells were cultured in DMEM medium (Sigma-Aldrich, United States, D6429) supplemented with 10% fetal bovine serum (FBS, Gibco, United States, A5669701) and 1% penicillin-streptomycin (Sigma-Aldrich, United States, P7539) at 37°C under 5% $CO_2$. MO3.13 cells were cultured in DMEM medium containing 10% FBS and 1% penicillin-streptomycin. SH-SY5Y cells were cultured in DMEM medium with 10% FBS and 1% penicillin-streptomycin. Prior to differentiation induction, SH-SY5Y cells were adapted for 2 days. Once cells reached 60–70% confluence, they were treated with 10 µM all-trans retinoic acid (RA, Sigma-Aldrich, United States, R2625) for 48–72 hours to induce differentiation into a neuron-like state.

**MS in *vitro* model induction.** To establish a co-culture model comprising human microglial cells, oligodendrocytes, and neurons to simulate the pathological conditions of MS in the CNS *in vitro*, HMC3 cells were first cultured in the upper chamber of a transwell plate, while MO3.13 and SH-SY5Y cells were co-cultured at a 1:2 ratio in the lower chamber. Subsequently, the medium was replaced with one containing 0.5 mg/mL lysophosphatidylcholine

(LPC, Merck, Germany, 1372050) for 24 hours to chemically induce oligodendrocyte injury, mimicking the demyelination process observed in MS *in vivo* [30,31]. Microglial cells were utilized to sense and simulate the neuroinflammatory microenvironment in MS.

## Genetic manipulation

**Oligodendrocyte overexpression *TRIM37*.** To investigate the function of TRIM37 in oligodendrocytes, the full-length coding sequence of human *TRIM37* was cloned into the ViraPower lentiviral expression vector (Thermo Fisher Scientific, United States, K495000) according to the manufacturer's standard protocol. MO3.13 cells were seeded in 6-well plates at a density of $1 \times 10^5$ cells per well. Once cells reached 50% confluence, 8 µg/mL Polybrene (Sigma-Aldrich, United States, S2667) was added to enhance transfection efficiency, followed by infection with overexpression *TRIM37* virus for 48 hours before replacing the medium with fresh culture medium. Subsequently, overexpression efficiency was assessed using qRT-PCR and Western blot analyses.

## Molecular analyses

**QRT -PCR.** QRT-PCR was employed to quantitatively detect the mRNA expression levels of target genes. Total RNA was extracted using TRIzol reagent (Thermo Fisher Scientific, United States, 15596018CN), and RNA concentration and purity were determined using a NanoDrop micro-spectrophotometer, with an A260/A280 ratio ranging from 1.8 to 2.1. One microgram of RNA template was reverse-transcribed into cDNA using the PrimeScript RT Reagent Kit (Takara, Japan, RR037Q). qPCR amplification was performed using TB Green Premix (Takara, Japan, RR820Q) with the following program: 95°C for 30 seconds, followed by 40 cycles of 95°C for 10 seconds and 60°C for 30 seconds. Primer sequences are provided in Table 1, with GAPDH or β-actin serving as internal reference genes for normalization. The relative expression levels of target genes were calculated using the $2^{-\Delta\Delta Ct}$ method, and all experiments were performed with six biological replicates to ensure data reliability.

**Western blot.** Western blot analysis was performed to detect the expression levels of target proteins. Cells or tissue samples were lysed using RIPA lysis buffer (Thermo Fisher Scientific, United States, 89900) containing protease inhibitors (1:100) and phosphatase inhibitors (1:100) on ice for 30 minutes. After centrifugation at 4°C and 12,000 rpm for 20 minutes, the supernatant was collected, and protein concentration was determined using the BCA protein assay kit (Thermo Fisher Scientific, United States, 23235) and adjusted to 1−5 mg/mL. Depending on the molecular weight of the target protein, an appropriate concentration of separating gel was selected. Thirty to fifty micrograms of protein samples were mixed with 5 × Laemmli buffer at a 1:4 ratio and denatured at 100°C for 5 minutes before being loaded onto SDS-PAGE gels for electrophoresis at 80V for the stacking gel and 120V for the separating gel. Following electrophoresis, the gel and PVDF membrane (Thermo Fisher Scientific, United States, 88518) were assembled into a transfer system, and proteins were transferred using the wet transfer method at 100 mA for 60−90 minutes. Transfer efficiency was verified by Ponceau S staining, followed by three washes with TBST. The membrane

**Table 1. Primer sequences and information.**

| Primers | Amplicon size | F/R | Sequence(5'-3') |
|---|---|---|---|
| *TRIM37* | 108 | Forward | CGGGCTAATGACCGAGAATAC |
| | | Reverse | CTGCTCAAAGAGGTTGACAGG |
| *IL1B* | 146 | Forward | GTTGCTGGTCACATTCCTGG |
| | | Reverse | GCAGGTAATCCCAAAAGCGAC |
| *TNF* | 220 | Forward | CCTCTCTCTAATCAGCCCTCTG |
| | | Reverse | GAGGACCTGGGAGTAGATGAG |

was then blocked with 5% skim milk at room temperature for 1 hour. Primary antibodies, including TRIM37 (rabbit polyclonal antibody, Proteintech, United States, 13037–1-AP, dilution 1:1000), PEX5 (rabbit polyclonal antibody, Proteintech, United States, 12545–1-AP, dilution 1:1000), myelin basic protein (MBP) (rabbit polyclonal antibody, Proteintech, United States, 10458–1-AP, dilution 1:2000), ubiquitin (rabbit polyclonal antibody, Proteintech, United States, 10201–2-AP, dilution 1:1000), and internal reference β-actin (mouse monoclonal antibody, Cell Signaling Technology, United States, 3700, dilution 1:5000), were incubated overnight at 4°C. After three washes with TBST, horseradish peroxidase (HRP)-conjugated secondary antibodies, including goat anti-rabbit IgG (Beyotime, China, A0208, dilution 1:1000) and goat anti-mouse IgG (Beyotime, China, A0216, dilution 1:1000), were incubated at room temperature for 1 hour in the dark. Following three washes with TBST, the membrane was developed using ECL chemiluminescence reagent (Thermo Fisher Scientific, United States, 32209), exposed using a gel imaging system, and analyzed for grayscale values using ImageJ. All experiments were performed with three biological replicates to ensure data reliability.

**Immunofluorescence.** Immunofluorescence was used to detect the localization and expression of specific proteins within cells. Adherent cells were seeded in 24-well plates and grown to 60%−70% confluence before being washed twice with PBS (Thermo Fisher Scientific, United States, 10010023). Subsequently, cells were fixed with 4% paraformaldehyde (Merck, Germany, 441244) at room temperature for 30 minutes, followed by three washes with PBS, each lasting 5 minutes. For intracellular antigen detection, cells were permeabilized with 0.1% Triton X-100 (Thermo Fisher Scientific, United States, HFH10) at room temperature for 10 minutes, followed by three washes with PBS. After permeabilization, cells were blocked with blocking buffer containing 5% BSA (Thermo Fisher Scientific, United States, AM2616) at room temperature for 1 hour to prevent non-specific binding. Primary antibodies, including PEX5 (rabbit polyclonal antibody, Proteintech, United States, 12545–1-AP, dilution 1:100) and TRIM37 (mouse monoclonal antibody, Santa Cruz Biotechnology, United States, sc-514828, dilution 1:200), were incubated overnight at 4°C in a humidified chamber. The following day, cells were washed three times with PBST (PBS containing 0.1% Tween-20), each lasting 5 minutes, followed by incubation with fluorescence-labeled secondary antibodies, including goat anti-rabbit IgG (Thermo Fisher Scientific, United States, A-11034, dilution 1:1000) and goat anti-mouse IgG (Thermo Fisher Scientific, United States, A-11005, dilution 1:1000), at room temperature for 1 hour in the dark. To minimize background interference, the washing steps were repeated five times, with three washes with PBST and two washes with distilled water. Finally, cells were stained with DAPI (Thermo Fisher Scientific, United States, 62248, dilution 1:4000) for 10 minutes to label cell nuclei, followed by three washes with ultrapure water. After mounting with anti-fade mounting medium, cells were observed and imaged under a fluorescence microscope, with fluorescence intensity analyzed using ImageJ software. All experiments were performed with three biological replicates to ensure data reliability.

**Co-immunoprecipitation (Co-IP).** To verify the effect of TRIM37 on the ubiquitination level of PEX5 protein, Co-IP technology combined with Western blot analysis was employed. Prior to the experiment, cells were washed with pre-chilled PBS and collected, followed by lysis in a lysis buffer containing protease inhibitors on ice for 30 minutes. After centrifugation at 4°C and 12,000 rpm for 10 minutes, the supernatant was collected, and protein concentration was determined. Five hundred to one thousand micrograms of total protein were incubated with a PEX5 antibody overnight at 4°C. Following incubation, the antibody-protein complex was captured by adding pre-coated Protein A/G magnetic beads to the centrifuge tube and incubating at 4°C with gentle rotation for 2–3 hours. After magnetic separation, the magnetic beads were washed 3–5 times with lysis buffer to remove non-specifically bound proteins. Bound proteins were then eluted using a low-pH elution buffer or by heating with SDS-containing sample buffer. Eluted samples were separated by SDS-PAGE, transferred to a membrane, and incubated with primary antibodies against ubiquitin or TRIM37 and secondary antibodies for detection. The presence of target proteins and ubiquitination bands was observed. To ensure experimental reliability, all experiments were performed with at least three biological replicates, and Input samples were used to verify the expression levels of target proteins.

   

## Functional assays

**Reactive oxygen species (ROS) detection.** Intracellular ROS levels were monitored using a ROS green fluorescence detection kit (Sigma-Aldrich, United States, MAK143). Cells were seeded in 24-well plates and grown to 50%−70% confluence before being washed twice with serum-free medium. The fluorescent probe was diluted to a final concentration of 10 μM at a 1:1000 ratio in serum-free medium and incubated with cells at 37°C for 20–30 minutes under dark conditions. Following incubation, cells were washed 2–3 times with serum-free medium to remove unincorporated probe. Subsequently, cells were fixed with 4% paraformaldehyde for 10–15 minutes and permeabilized with 0.1% Triton X-100 at room temperature for 5–10 minutes to enhance cell membrane permeability and facilitate probe reaction with ROS. After permeabilization, cells were washed three times with serum-free medium. To minimize non-specific binding, cells were blocked with blocking buffer containing 5% BSA at room temperature for 30 minutes. Fluorescence intensity was observed under a fluorescence microscope with an excitation wavelength of 488 nm and an emission wavelength of 525 nm. Fluorescence intensity was quantitatively analyzed using ImageJ software. All experiments were performed with three biological replicates to validate data reliability.

**TUNEL cell apoptosis detection.** Cell apoptosis levels were assessed using a one-step TUNEL detection kit (Thermo Fisher Scientific, United States, C10618) Cells were fixed with 4% paraformaldehyde at room temperature for 30–60 minutes, followed by permeabilization with PBS containing 0.1% Triton X-100 at room temperature for 5–10 minutes to enhance cell membrane permeability. Subsequently, cells were washed thoroughly with PBS to remove unbound reagents. TUNEL reaction mixture was prepared according to the manufacturer's instructions and incubated with cells at 37°C for 60 minutes under dark conditions to allow the labeling probe to bind to DNA fragmentation ends. Following incubation, cells were washed 3–5 times with PBS and counterstained with DAPI (1:1000 dilution) for 5 minutes to label cell nuclei, followed by three washes with ultrapure water or PBS. After mounting with anti-fade mounting medium, apoptotic cells were observed under a fluorescence microscope, with fluorescence intensity quantitatively analyzed using ImageJ software. To ensure experimental reliability, all experiments were performed with at least three biological replicates, and strict control of fixation, permeabilization, and reaction times was implemented to avoid false-positive or false-negative results.

**CHX method for protein half-life detection.** To assess the stability of the PEX5 protein, cycloheximide (CHX) (Thermo Fisher Scientific, United States, SR0222C) treatment combined with Western blot technology was employed to determine its half-life. Cells were cultured in complete medium containing 10% fetal bovine serum until they reached 70%−80% confluence. Subsequently, the medium was replaced with serum-free medium for a 24-hour adaptation period to stabilize the protein synthesis background. The experimental group received a final concentration of 100 μM CHX to inhibit new protein synthesis, while a control group without CHX was also established. Cells were collected at 0, 1, 2, and 4 hours, washed with pre-chilled PBS, and lysed using RIPA lysis buffer containing protease inhibitors. After centrifugation at 12,000 rpm for 10 minutes at 4°C, the supernatant was collected for protein concentration determination. Equal amounts of total protein (50 μg) were separated by SDS-PAGE, transferred to a membrane, and subjected to Western blot analysis using a specific primary antibody against PEX5 (diluted 1:1000) and β-actin as an internal control (diluted 1:1000). The grayscale values of the target protein bands were quantified using ImageJ software. Using the protein expression levels of the group without CHX as a baseline, a degradation curve of the PEX5 protein over time was plotted. To ensure experimental reliability, strict control of CHX concentration, treatment duration, and lysis conditions was necessary, and three biological replicates were performed.

## In *vivo* validation

**EAE model establishment.** Eighteen 6–8-week-old male C57BL/6 mice, weighing 18–22 g, were purchased from Beijing Spf Biotechnology Co., Ltd. All animal experimental protocols were reported according to the ARRIVE guidelines (https://arriveguidelines.org) and reviewed and approved by the Animal Ethics Committee of Beidahuang

Group General Hospital (approval number: KY2024092003) The experimental mice were housed in an SPF-grade animal room under constant temperature and humidity conditions, with free access to food and water. After one week of adaptive feeding, the mice were randomly divided into three groups: a blank control (Con) group, an experimental autoimmune encephalomyelitis (EAE) group, and an EAE+TRIM37 group (overexpressing *TRIM37*), with six mice in each group. To establish the EAE model, 200 µL of an emulsion containing 100 µg/mL myelin oligodendrocyte glycoprotein (MOG35–55) (Sigma-Aldrich, United States, M4939) and an equal volume of complete Freund's adjuvant (CFA) was injected subcutaneously into the axillary, inguinal, and dorsal regions of the mice on day 0. Subsequently, 200 ng of pertussis toxin (PTX) (Sigma-Aldrich, United States, P7208) was injected intraperitoneally on days 0 and 2 to enhance the immune response. The control group received an equal volume of saline in the same locations. For overexpression *TRIM37*, AAV2/9 viruses carrying *TRIM37* cDNA were injected into the corpus callosum region at a dose of 0.5–1 µL per side. All mice were anesthetized via intraperitoneal injection of 30 mg/kg sodium pentobarbital on the 14th day after immunization. Subsequently, the mice were placed in a sealed container, and $CO_2$ was slowly introduced, with the concentration gradually increased to 80% over a period of 5 minutes. This was done to ensure that the mice gradually lost consciousness and experienced respiratory depression. When apnea, mydriasis, and absence of toe-pinch reflex were observed, cervical dislocation was immediately performed to ensure rapid death of the mice. Thereafter, the corpus callosum tissues were harvested and subjected to HE staining, LFB (luxol fast blue) staining, and immunohistochemistry to verify demyelination and inflammatory infiltration in the central nervous system.

## Histopathological staining

**HE staining.** Corpus callosum samples were fixed with 4% paraformaldehyde and processed using the paraffin embedding method. Sections were cut to a thickness of 4–6 µm and then immersed in xylene I and II for 10 minutes each to dissolve the paraffin, followed by hydration through two 5-minute washes in absolute ethanol and 5-minute washes in 95%, 85%, and 70% gradient ethanol. After hydration, the sections were rinsed with distilled water and stained with hematoxylin for 10 minutes, followed by rinsing with running water and differentiation with 1% hydrochloric acid-ethanol solution for 5 seconds. Bluing was performed in warm water or a 0.75% ammonium chloride weak alkaline solution to enhance nuclear staining. Subsequently, the sections were transferred to eosin stain for 5 minutes, dehydrated through 3-minute washes in 70%, 85%, 95%, and absolute ethanol, and cleared in xylene I and II for 5 minutes each. Finally, the sections were mounted with neutral balsam to avoid air bubbles interfering with observation. The entire process required strict control of time and reagent concentration to ensure that the nuclei appeared vivid blue and the cytoplasm and collagen fibers were stained pink to peach, clearly presenting the histological features of the tissue. Three biological replicates were performed for each experimental group to verify data reliability.

**LFB staining.** After hydration of the aforementioned paraffin sections, the sections were placed in 0.1% LFB stain (Beyotime, China, C0631S) and incubated overnight in a 60°C oven, followed by rinsing with tap water to remove excess stain. Subsequently, the sections were immersed in 70% ethanol for initial differentiation and alternately differentiated with 0.05% lithium carbonate solution and 70% ethanol under microscopic observation until the boundary between gray matter and white matter was clear, the gray matter color faded, the myelin sheath appeared bright blue, and the background was colorless or light blue. After differentiation, the sections were rinsed with distilled water, counterstained with 0.1% eosin for 1 minute to enhance cytoplasmic contrast, and then dehydrated through 5-minute washes in 70%, 80%, 95%, and absolute ethanol, and cleared in xylene I and II for 5 minutes each. Finally, the sections were mounted with neutral balsam to avoid air bubbles interfering with observation. The entire process required strict control of the differentiation time to ensure the specificity of myelin staining. Additionally, counterstaining could be followed by n-butanol dehydration to optimize the staining effect.

**Immunohistochemistry.** Tissue samples were fixed with 4% paraformaldehyde and prepared as 4–5 µm sections using the paraffin embedding method. The sections were attached to poly-L-lysine-pretreated slides and baked at 60°C

for 30–60 minutes to enhance adhesion. The sections were hydrated through xylene and gradient ethanol to distilled water, followed by inactivation of endogenous peroxidase with 3% $H_2O_2$-PBS solution for 10 minutes and three washes with PBS. Heat-induced antigen retrieval was performed using citrate buffer (pH 6.0) heated to boiling and maintained for 10 minutes. After blocking non-specific binding, diluted MBP primary antibody (rabbit-derived antibody, Abcam, ab40390, dilution ratio: 1:200) was added and incubated overnight at 4°C, followed by incubation with HRP-labeled secondary antibody (goat anti-rabbit IgG, Beyotime, A0208, dilution ratio 1:200) at room temperature for 30–60 minutes. DAB color development working solution was used for 5 minutes of darkroom color development, and hematoxylin was used to counterstain the nuclei, followed by gradient dehydration and clearing. Finally, the sections were mounted with neutral balsam for observation. Three biological replicates were performed for each experimental group to verify data reliability.

### Data statistics

Experimental data were analyzed and processed using GraphPad Prism 10.0 software. For normally distributed data, intergroup comparisons were performed using an independent samples t-test (for two groups) or one-way ANOVA (for more than two groups). If ANOVA showed significant differences, Tukey's HSD was used for multiple comparison correction. For non-normally distributed data, intergroup comparisons were performed using the Mann-Whitney U test (for two groups) or Kruskal-Wallis H test (for more than two groups), with Dunn's method for subsequent multiple comparison correction. Count data were analyzed using the chi-square test or Fisher's exact test. All data were presented as mean±standard error of the mean (Mean±SEM), and $P < 0.05$ was considered statistically significant. In the figures, $*P < 0.05$, $**P < 0.01$, $***P < 0.001$, and $****P < 0.0001$ to visually demonstrate the significance level of the statistical results.

## Results

### Differential expression gene (DEG) analysis and weighted gene co-expression network analysis (WGCNA) in MS patient tissues

We first performed DEG analysis on 40 MS patient samples and 10 control samples from the GSE135511 dataset to screen for potential therapeutic targets for MS. Using a threshold of |log2FC| ≥ 1 and adjusted p-value<0.05 (Fig 1a), a total of 97 significant DEGs were identified, with 43 genes downregulated and 54 genes upregulated in the MS group. Network topology analysis (Figs 1b and 1c) indicated that the constructed weighted gene co-expression network conformed to the characteristics of a scale-free network. To gain a deeper understanding of the key genes associated with MS and their mutual relationships, we employed WGCNA to identify the blue gene module significantly associated with MS lesions (Figs 1d and 1e). By setting criteria for module membership (MM > 0.8) and gene significance (GS > 0.5), 450 core genes were precisely extracted from the blue module (Fig 1f). These core genes occupy key positions in the co-expression network, and their expression changes may synergistically influence the disease progression of MS.

### Identification of key genes based on enrichment analysis, machine learning, and PPI network

To further screen key genes closely related to MS, we conducted an intersection analysis between the previously identified DEGs and the significant module genes from the WGCNA analysis, ultimately obtaining 46 candidate key genes (Fig 2a). GO functional enrichment analysis of these 46 genes revealed their involvement in multiple important biological processes, including ubiquitin-protein ligase binding, mitochondrial respiratory chain complex, and macroautophagy (Fig 2b). KEGG pathway enrichment analysis further indicated that these genes were significantly enriched in pathways related to various neurodegenerative diseases, including Alzheimer's disease, Huntington's disease, and amyotrophic lateral sclerosis (Fig 2c), suggesting that these pathways may play important roles in the pathogenesis of MS. To further identify key regulatory factors from the candidate genes, the random forest algorithm was applied for feature gene screening. Based on the Gini index ranking, *TRIM37* exhibited a high discriminative ability, indicating its potential key role in the

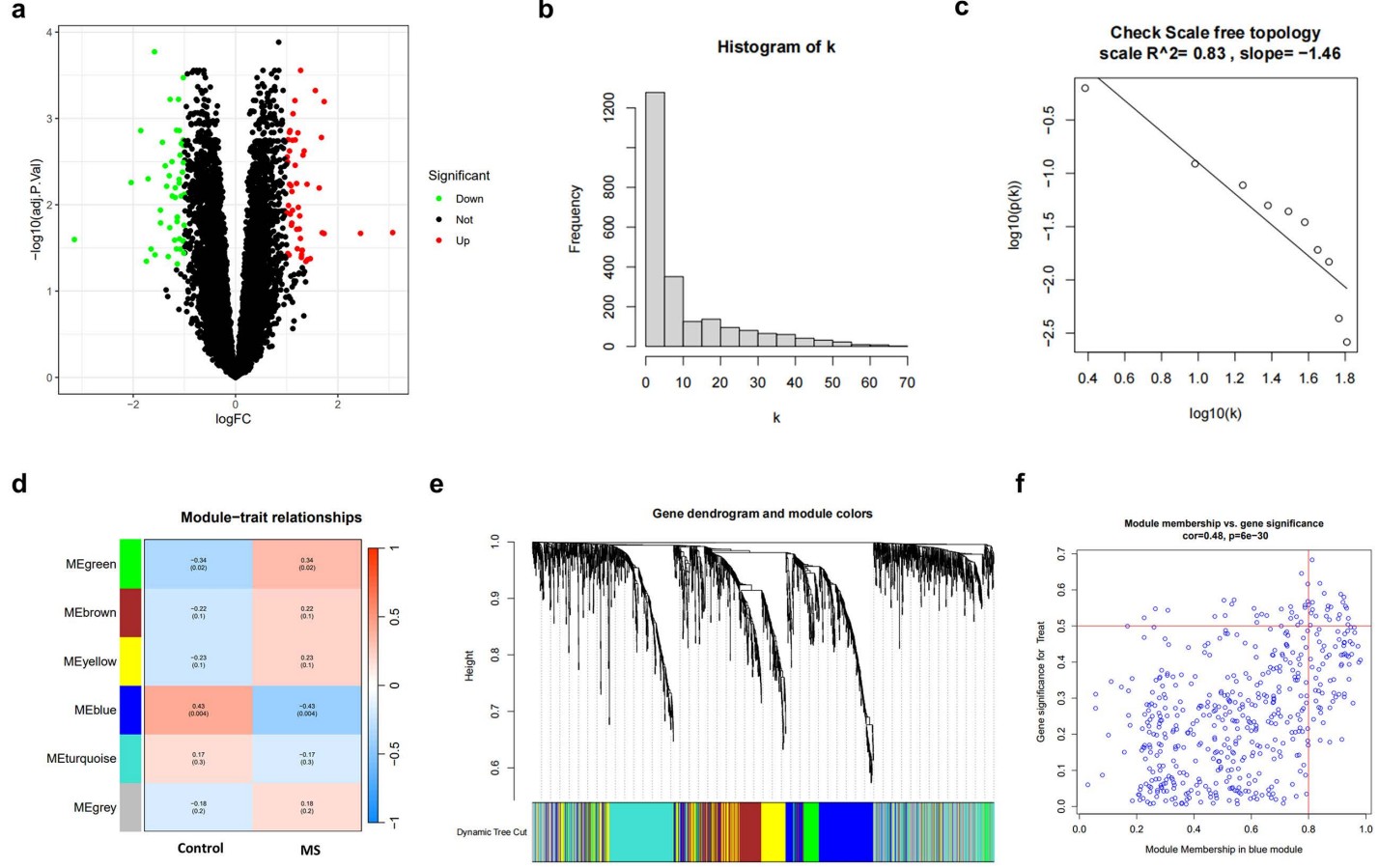

**Fig 1. Identification of gene co-expression modules associated with MS.** **(a)** Volcano plot showing DEGs between MS patients and controls. **(b)** Histogram of gene connectivity (k) distribution, reflecting the number of genes with different connectivity degrees in the network. **(c)** Test plot for verifying whether the network conforms to scale-free topology. **(d)** Module correlation matrix heatmap, where each module represents a set of genes, and the color indicates the correlation between the module and a given trait. **(e)** Gene dendrogram based on hierarchical clustering and its module assignment, with different color bars representing different co-expression modules. **(f)** Relationship curve between average connectivity and soft-thresholding power exponent, used to determine the optimal power exponent.

occurrence and development of MS (Fig 2d). It is worth noting that *SLC5A3*, which ranked first, has already been extensively studied regarding its functions. Therefore, we ultimately chose to study *TRIM37*. Subsequently, the STRING database was used to construct a PPI network for TRIM37 (confidence score>0.7), revealing tight interactions with multiple functionally related proteins, such as TRAF6, RBCK1, and PEX5 (Fig 2e). This suggests that *TRIM37* may play a central regulatory role in the pathogenesis of MS by modulating various biological processes.

## Downregulation of TRIM37 expression in the MS model is accompanied by exacerbated neuroinflammation, oxidative stress, and cell apoptosis

In this study, we successfully established a transwell co-culture system based on three cell types (neurons-oligodendrocytes-microglia) and used LPC to induce an *in vitro* MS model, as detailed in the methods section. Western blot analysis showed that the expression level of MBP (mainly from oligodendrocytes) in the lower chamber of the transwell system was significantly lower in the LPC-treated group than in the control group (*P*<0.01) (Figs 3a and 3b, S1 Fig in S1 File), indicating that

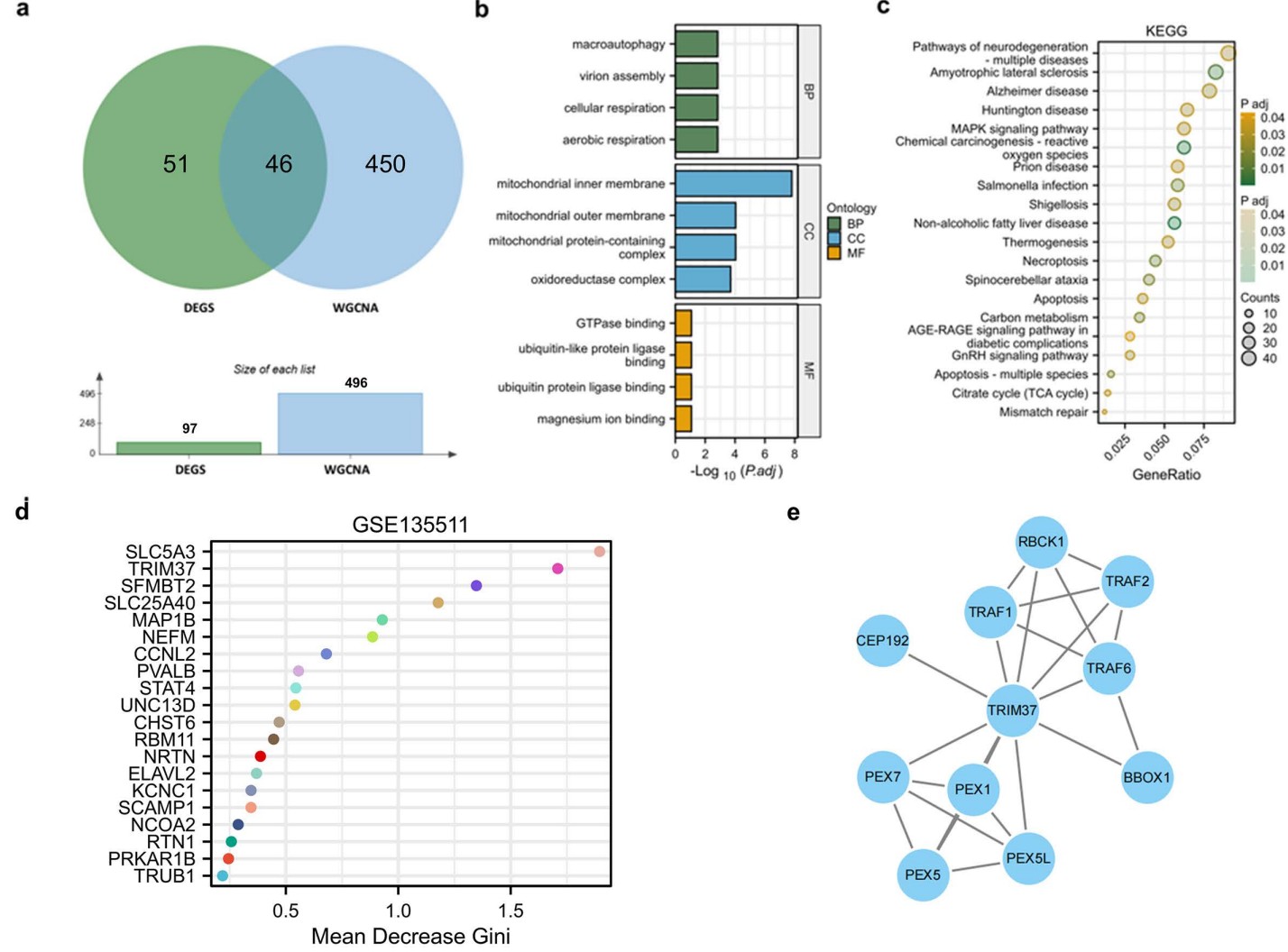

**Fig 2. Enrichment analysis of genes from the significant MS module. (a)** Venn diagram showing the intersection of DEGs and significant module genes from WGCNA, resulting in 46 key intersection genes. **(b)** GO functional enrichment analysis results, including three dimensions: biological process (BP), cellular component (CC), and molecular function (MF). **(c)** KEGG pathway enrichment analysis results. The color of the dots represents the adjusted P-value (P.adj), and the size represents the number of genes. **(d)** Top 20 important feature genes identified by the random forest model, ranked by the average decrease in Gini index. **(e)** PPI network constructed using the STRING database, showing potential interactions among key genes.

the MS model induced by LPC can mimic the pathological features of demyelination. In terms of inflammatory response, qRT-PCR analysis of microglia in the upper chamber of the transwell system revealed a significant upregulation of the mRNA expression levels of pro-inflammatory cytokine genes *IL1B* and *TNF* in the model group compared to the control group (Fig 3c, *P*<0.0001), suggesting typical neuroinflammatory responses in the model group. Further studies showed that the ROS levels in oligodendrocytes and neurons in the lower chamber of the co-culture system in the model group were significantly increased (*P*<0.0001, Figs 3d and 3e), while TUNEL staining results indicated a significant increase in apoptosis levels in both cell types (*P*<0.0001, Figs 3f and 3g). Notably, qRT-PCR (Fig 3h) and Western blot analyses revealed a significant downregulation of the mRNA and protein expression levels of *TRIM37* in microglia in the model group (*P*<0.01, Figs 3i and 3j, S2 Fig in S1 File), suggesting that *TRIM37* may be involved in the occurrence and development of MS.

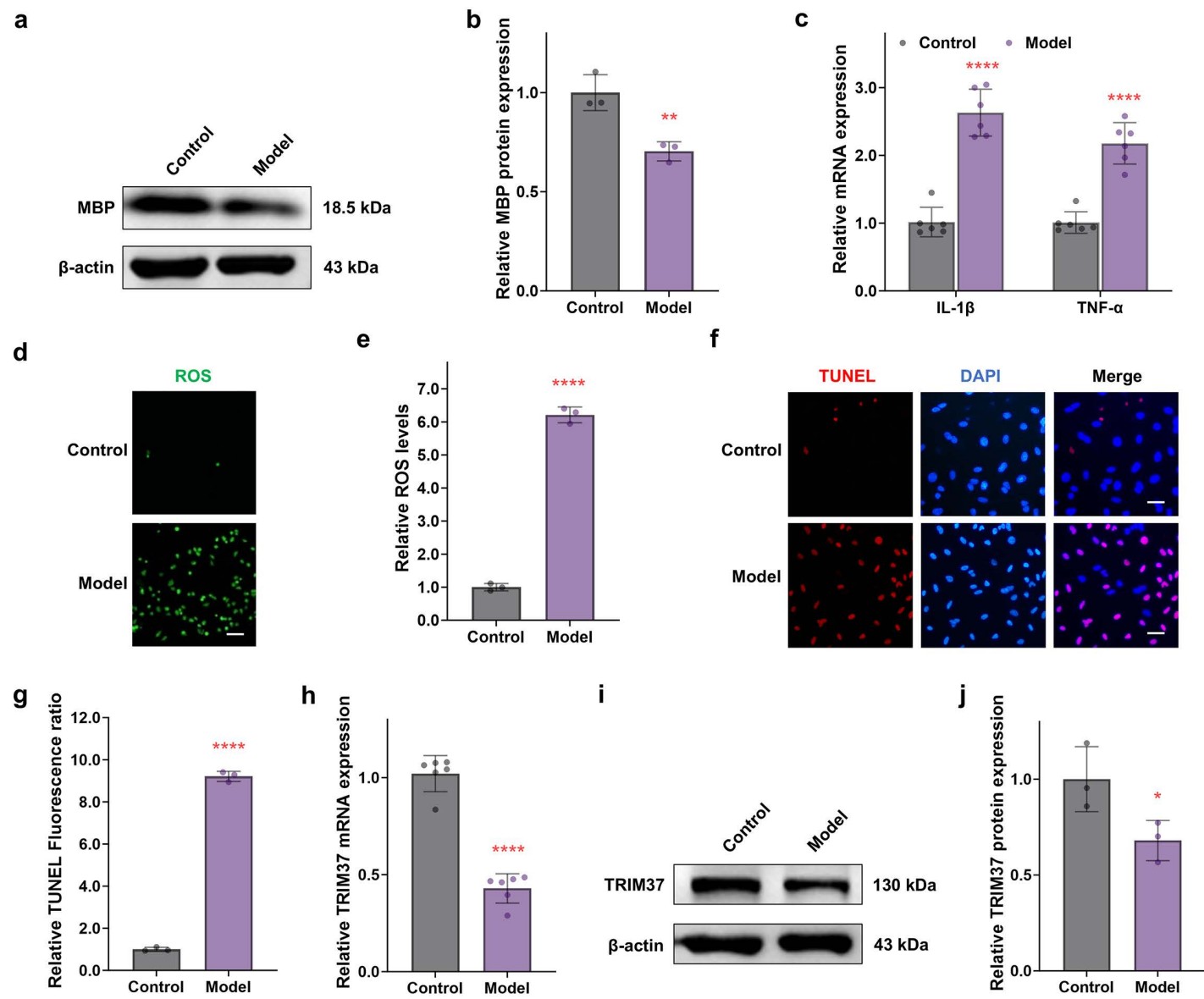

**Fig 3. Downregulation of TRIM37 expression exacerbates pathological damage in the MS model. (a, b)** Western blot analysis revealed a significant decrease in MBP expression in oligodendrocytes in the LPC-induced MS model group (n = 3) compared to the control group. (c) qRT-PCR results showed a significant increase in the mRNA expression levels of the pro-inflammatory cytokines *IL1B* and *TNF* in microglia in the MS model group (n = 6) compared to the control group. **(d, e)** ROS detection (n = 3) indicated a significant increase in intracellular oxidative stress levels in the model group compared to the control group. Scale bar = 100 μm. **(f, g)** TUNEL staining (n = 3) demonstrated a significant increase in the apoptosis rate in the model group compared to the control group. Scale bar = 50 μm. (h) qRT-PCR (n = 6) and **(i, j)** Western blot (n = 3) both showed a significant decrease in the *TRIM37* mRNA and protein levels in the LPC-induced MS model compared to the control group. * $P < 0.05$, ** $P < 0.01$, *** $P < 0.001$, **** $P < 0.0001$, ns = not significant.

## Overexpression *TRIM37* ameliorates oligodendrocyte Injury by promoting PEX5 monoubiquitination

To further investigate the regulatory mechanism of TRIM37 in oligodendrocytes in the MS model, we established a overexpression *TRIM37* system (OE-TRIM37) in oligodendrocytes and confirmed significant increases in both mRNA and protein expression levels in the overexpression *TRIM37* group compared to the LPC group using qRT-PCR (Fig 4a) and Western

blot (Figs 4b and 4c, S3 Fig in S1 File) (*P*<0.0001). Further Co-IP experiments demonstrated that overexpression *TRIM37* significantly enhanced the ubiquitination level of the PEX5 protein (Figs 4d and 4e, S4 Fig in S1 File, *P*<0.001). Additionally, colocalization experiments confirmed by immunofluorescence revealed a direct interaction between TRIM37 and PEX5 proteins (Fig 4f), suggesting that TRIM37 may regulate PEX5 protein expression through ubiquitination modification. CHX protein stability experiments showed that overexpression *TRIM37* significantly delayed the degradation rate of the PEX5 protein (Figs 4g and 4h, S5 Fig in S1 File), indicating that maintaining PEX5 protein stability through the monoubiquitination pathway is an important mechanism by which TRIM37 exerts its neuroprotective effects.

## Role of the TRIM37-PEX5 signaling axis in ameliorating myelin damage

This study further elucidated the protective mechanism of the TRIM37-PEX5 signaling pathway in MS. Western blot results showed that overexpression *TRIM37* in oligodendrocytes significantly upregulated the protein expression level of PEX5 (*P*<0.001, Figs 5a and 5b, S6 Fig in S1 File), indicating a positive regulatory effect of TRIM37 on PEX5. ROS detection experiments revealed that overexpression *TRIM37* effectively reduced oxidative stress levels (*P*<0.001, Figs 5c and 5d), suggesting that overexpression *TRIM37* can restore peroxisome function and reduce ROS accumulation by enhancing PEX5 stability. TUNEL staining analysis indicated that overexpression *TRIM37* significantly inhibited apoptosis in oligodendrocytes (*P*<0.001, Figs 5e and 5f). More importantly, the expression level of the key myelin protein MBP was significantly increased in the overexpression *TRIM37* group (*P*<0.01, Figs 5g and 5h, S7 Fig in S1 File), confirming that TRIM37 promotes myelin formation and repair through a PEX5-dependent mechanism. These findings provide new molecular insights into the neuroprotective role of TRIM37 in MS and lay a theoretical foundation for the development of MS treatment strategies targeting the TRIM37-PEX5 axis.

## In *vivo* validation of overexpression *TRIM37* in alleviating MS and demyelination

To validate the myelin-protective effect of TRIM37 in MS, we established an EAE mouse model to investigate the *in vivo* effects of overexpression *TRIM37* on neuroinflammatory infiltration and myelin repair. HE staining results (Fig 6a) showed significant inflammatory cell infiltration in the corpus callosum region of EAE group mice, reflecting typical neuropathological features of MS. Importantly, overexpression *TRIM37* reduced inflammatory cell infiltration, suggesting a neuroprotective effect of TRIM37 in inhibiting neuroinflammation. qRT-PCR detection of corpus callosum tissue (Figs 6b and 6c) further revealed a significant upregulation of the mRNA levels of the pro-inflammatory cytokines *IL1B* and *TNF* in the EAE group compared to the Con group (*P*<0.001), while overexpression *TRIM37* significantly downregulated the expression of these inflammatory factors compared to the EAE group (*P*<0.001), suggesting that TRIM37 may indirectly affect the neuroinflammatory microenvironment by regulating microglia. In terms of myelin repair, LFB staining results showed a decrease in myelin blue staining in the EAE group compared to the Con group (Fig 6d), indicating widespread demyelination, while overexpression *TRIM37* significantly alleviated the degree of myelin damage (*P*<0.001, Fig 6e). Immunohistochemical staining of MBP in corpus callosum tissue showed a significant decrease in MBP expression in the EAE group (*P*<0.05, Figs 6f and 6g), but overexpression *TRIM37* restored MBP expression levels (*P*<0.001). Collectively, these results indicate that overexpression *TRIM37* not only inhibits inflammatory cell infiltration in the EAE model but also alleviates demyelination, further supporting its protective role in the development of MS.

## Discussion

This study, for the first time, unveils the mechanism by which TRIM37 regulates the stability of PEX5 through non-degradative ubiquitination, thereby ameliorating antioxidant imbalance and demyelination lesions in oligodendrocytes. In MS, low expression of TRIM37 leads to accelerated degradation of PEX5 protein, dysfunction of peroxisomes, subsequently triggering oxidative stress and cell apoptosis. However, TRIM37, by mediating non-degradative monoubiquitination

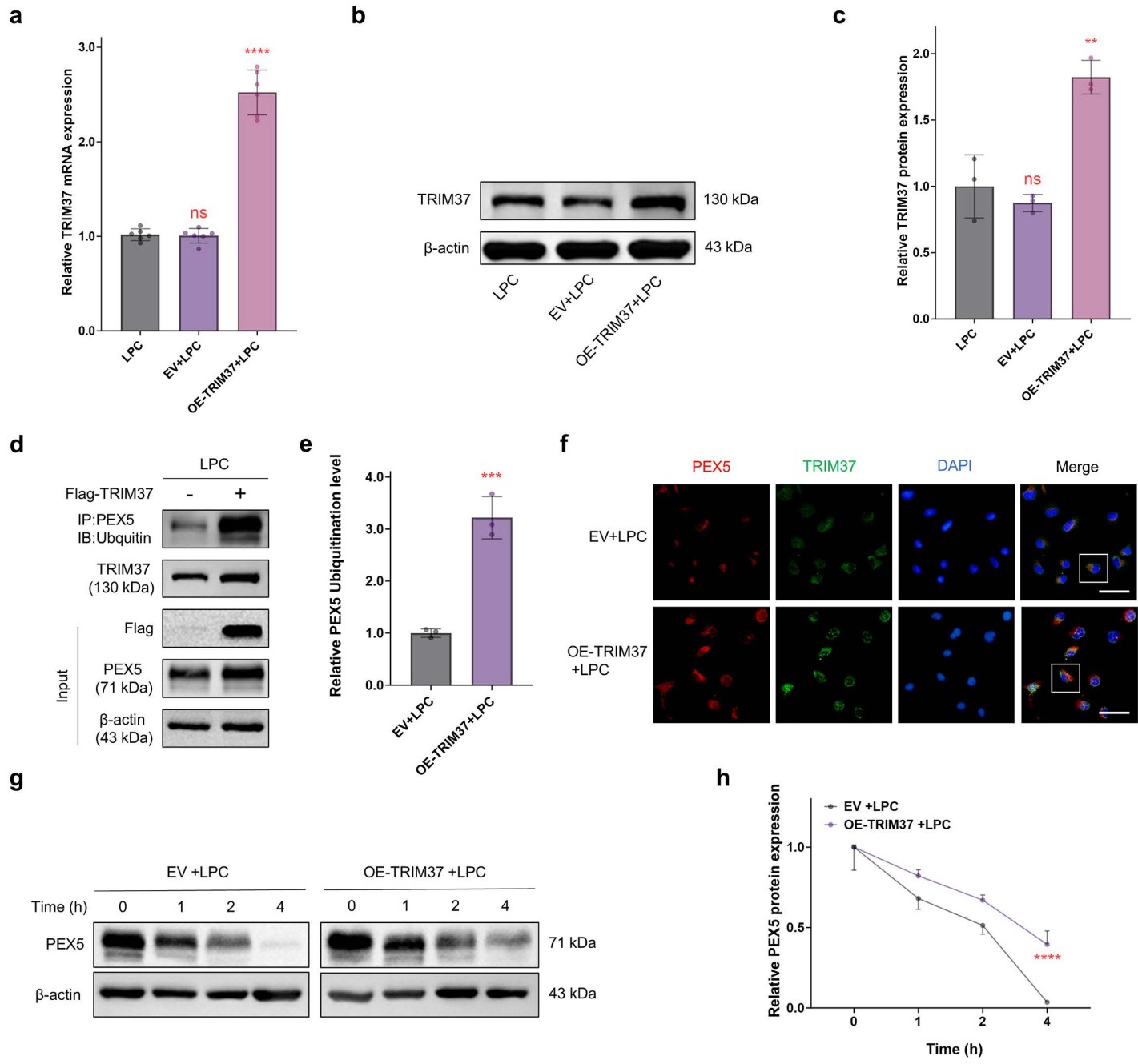

**Fig 4. Overexpression *TRIM37* enhances PEX5 protein stability by promoting PEX5 monoubiquitination.** (a) qRT-PCR (n = 6) and **(b, c)** Western blot (n = 3) results showed significant increases in the transcription and translation levels of TRIM37 in the overexpression *TRIM37* group (OE-TRIM37) compared to the LPC group. **(d, e)** Co-IP experiments (n = 3) revealed a significant enhancement of PEX5 ubiquitination modification in the OE-TRIM37 group compared to the LPC group. **(f)** Immunofluorescence colocalization detection (n = 3) showed overlapping regions of TRIM37 and PEX5 within cells, as indicated by the boxes in the figure. Scale bar = 50 μm. **(g, h)** CHX protein half-life experiments (n = 3) demonstrated a significantly lower degradation rate of PEX5 in the OE-TRIM37 group compared to the control group. * $P < 0.05$, ** $P < 0.01$, *** $P < 0.001$, **** $P < 0.0001$, ns = not significant.

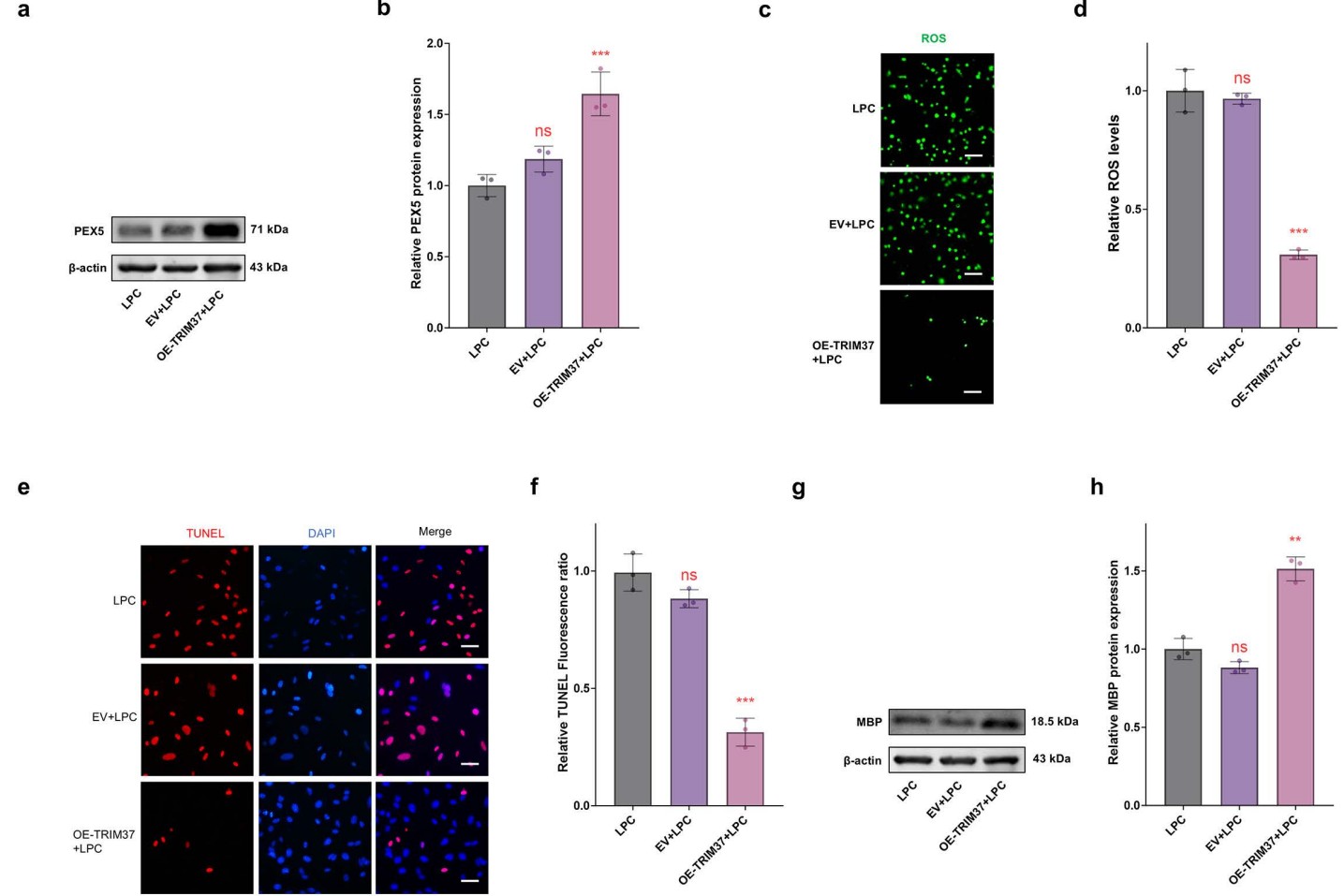

**Fig 5. Overexpression *TRIM37* promotes myelin repair and inhibits cell apoptosis by upregulating PEX5. (a, b)** Western blot (n = 3) showed a significant increase in PEX5 protein levels in the OE-TRIM37 group compared to the LPC group. **(c, d)** ROS fluorescence detection (n = 3) indicated a significant decrease in oxidative stress levels in the OE-TRIM37 group compared to the LPC group. Scale bar = 100 µm. **(e, f)** TUNEL apoptosis analysis (n = 3) showed a significant decrease in the apoptosis rate in the OE-TRIM37 group compared to the LPC group. Scale bar = 50 µm. **(g, h)** Western blot results (n = 3) confirmed a significant upregulation of MBP protein levels in the OE-TRIM37 group compared to the LPC group. * $P < 0.05$, ** $P < 0.01$, *** $P < 0.001$, **** $P < 0.0001$, ns = not significant.

modification of PEX5, enhances its stability, thereby restoring peroxisomal function, reducing ROS accumulation, inhibiting oligodendrocyte apoptosis, and promoting the expression of myelin basic protein (MBP) and myelin repair (Fig 7).

As an E3 ubiquitin ligase, the role of TRIM37 transcends the traditional stereotype of ubiquitin-mediated protein degradation. Conventionally, ubiquitination is commonly perceived to mediate protein degradation [32], yet certain E3 ubiquitin ligases can regulate substrate protein function through atypical ubiquitination modes, rather than promoting their degradation [33]. Specifically, monoubiquitination modification can finely tune protein stability and functional activity by altering protein conformation, interactions, or subcellular localization [33]. Against this backdrop, our study, combining Co-IP and CHX protein stability assays, discovered that TRIM37 specifically performs non-degradative monoubiquitination modification on PEX5, which not only avoids PEX5 degradation but instead enhances its stability. Importantly, the regulatory role of the TRIM37-PEX5 monoubiquitination axis may hold special significance in neurological diseases: on one hand, this

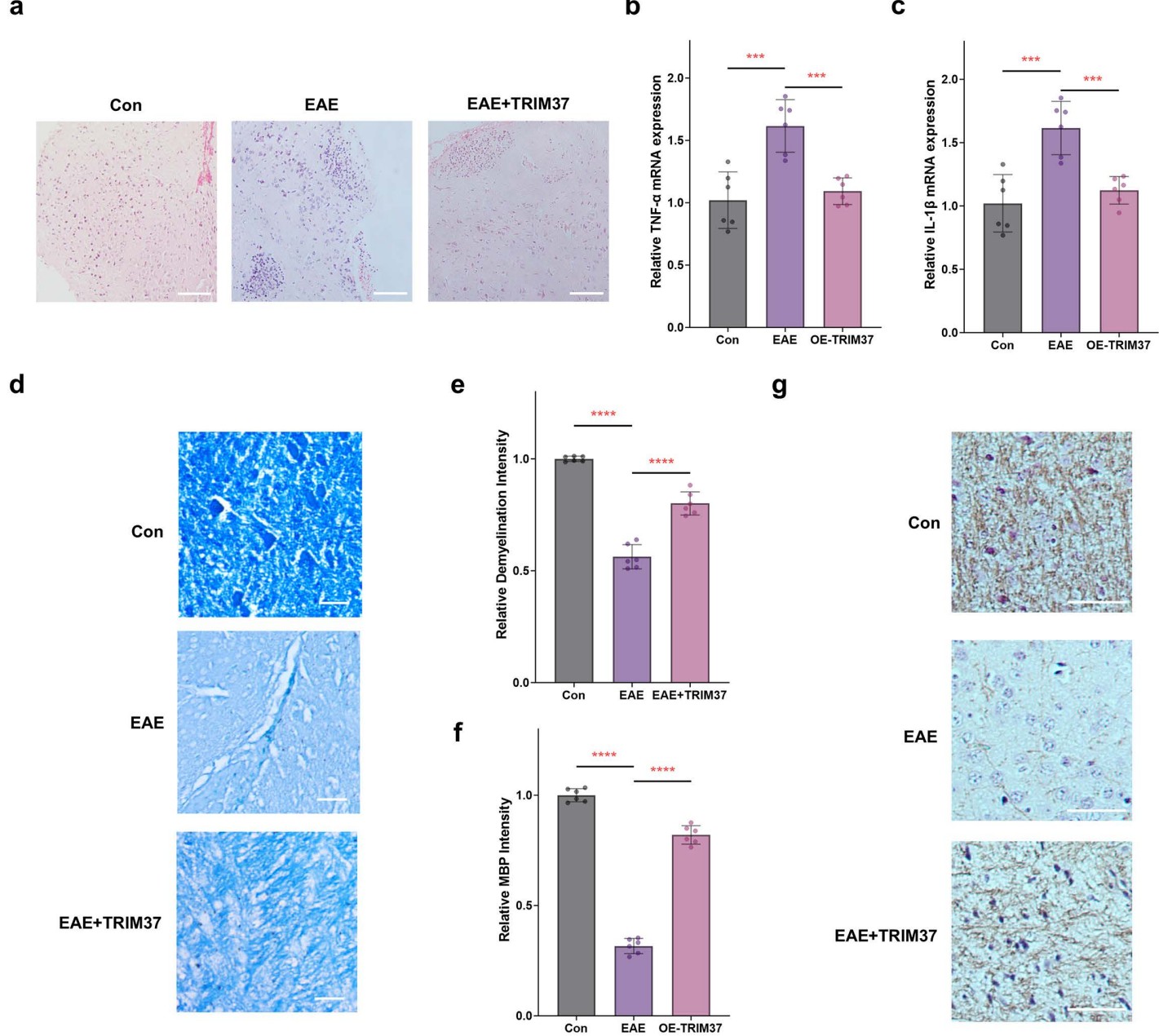

**Fig 6. Overexpression *TRIM37* inhibits inflammatory infiltration and alleviates demyelination in EAE model mice. (a)** HE staining results of corpus callosum tissue in mice. Scale bar = 100 μm. (b, c) qRT-PCR results of the mRNA levels of the inflammatory factors *TNF* and *IL1B*. **(d, e)** LFB staining and statistical graph of corpus callosum tissue in mice. Scale bar = 50 μm. **(f, g)** Immunohistochemical staining and statistical graph of MBP in corpus callosum tissue. Scale bar = 100 μm. Each group had six biological replicates for the above results. * $P < 0.05$, *** $P < 0.001$, **** $P < 0.0001$.

modification directly affects PEX5 stability; on the other hand, this mechanism may serve as a potential target for oxidative stress regulation in the nervous system.

Further analysis indicates that peroxisomal dysfunction has been extensively reported to be closely associated with various neurological diseases, such as Zellweger syndrome and adrenoleukodystrophy [27,28], but its specific role in

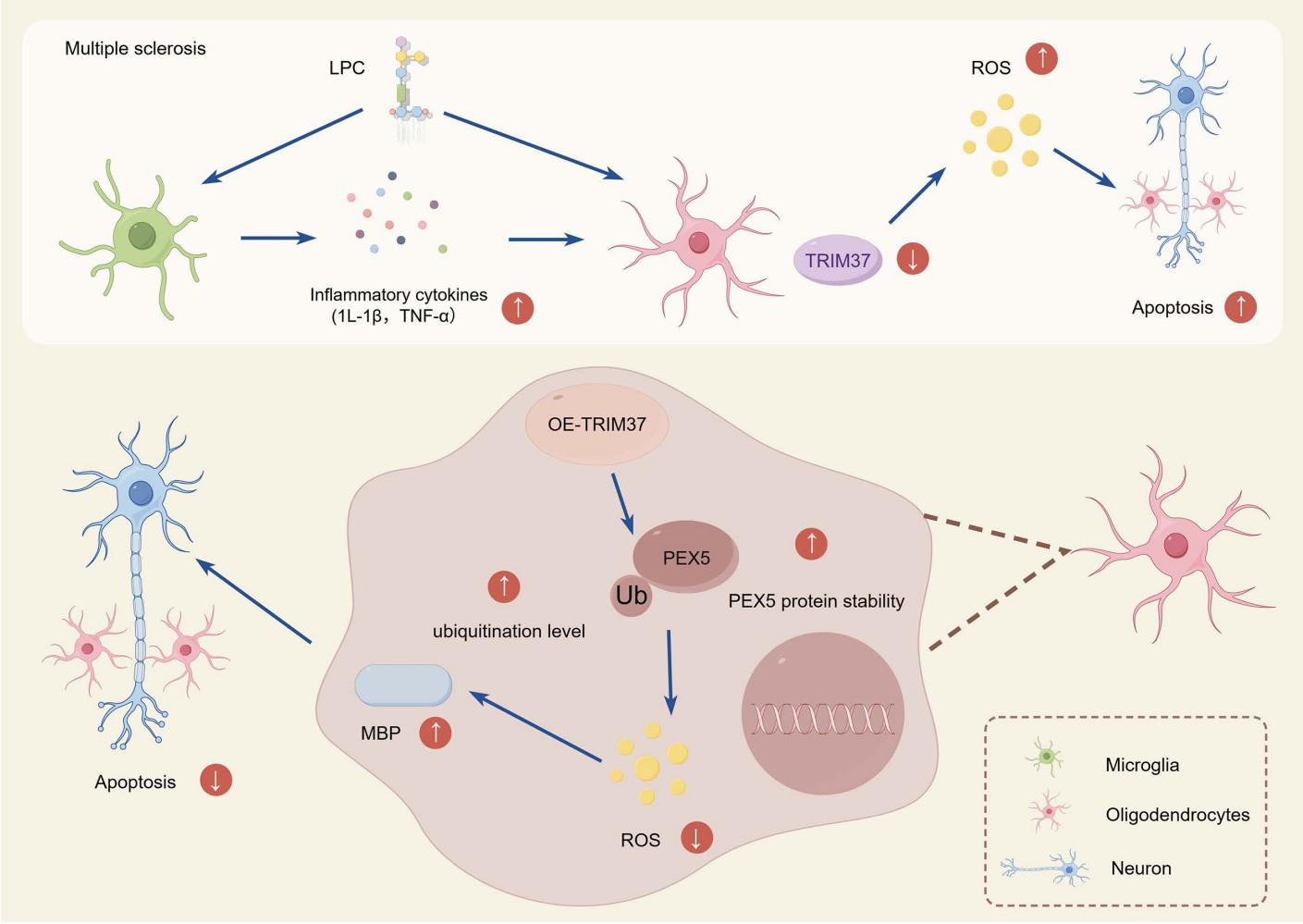

**Fig 7. Schematic diagram of the molecular mechanism by which TRIM37 improves MS by stabilizing PEX5 through monoubiquitination.** The upper part of the figure depicts the development mechanism of MS: LPC first stimulates microglia to release inflammatory cytokines IL-1β and TNF-α, forming an inflammatory microenvironment. This leads to a significant downregulation of *TRIM37* expression in oligodendrocytes, resulting in a large accumulation of ROS and ultimately triggering oxidative stress-induced apoptosis in oligodendrocytes and neurons. The lower part of the figure depicts the protective mechanism of overexpression *TRIM37* in MS: overexpression *TRIM37* (OE-TRIM37) in oligodendrocytes can reverse the above pathological process. TRIM37 enhances PEX5 protein stability through monoubiquitination modification. Stable PEX5 maintains the normal function of peroxisomes, clears ROS, and alleviates oxidative stress damage.

MS remains unclear. A significant advancement of this study is the revelation of the pivotal role of the TRIM37-PEX5-peroxisomal function axis in MS: TRIM37 stabilizes PEX5 through monoubiquitination, restoring peroxisomal activity, thereby alleviating oxidative stress damage and promoting myelin regeneration. This finding aligns with the study by Wang et al., which demonstrated that TRIM37 localizes to peroxisomes and regulates PEX5-dependent protein import [29], but our study is the first to extend this mechanism to the pathophysiological study of demyelinating diseases. In summary, our work not only elucidates the causal link between peroxisomal dysfunction and myelin damage in MS at the molecular level but also provides a novel metabolic regulatory perspective on the pathological mechanisms of MS.

Overexpression *TRIM37* strategy may offer a potential direction for antioxidant therapy in MS. The TRIM37-PEX5 axis mechanism could complement existing MS treatment approaches and warrants further investigation as a possible

component of combination therapy. Our study demonstrated in the EAE model that overexpression *TRIM37* improved myelin regeneration in spinal cord tissue, reduced levels of inflammatory cytokines (such as IL-1β and TNF-α), and decreased neurological function scores. These results suggest that TRIM37 might not only protects oligodendrocytes but also alleviates neuroinflammation by modulating microglia. These findings provide preliminary *in vivo* evidence of the dual role of the TRIM37-PEX5 axis in EAE models, though its clinical translatability requires further validation.

Despite revealing the critical role of the TRIM37-PEX5 axis in MS through bioinformatics analysis, *in vitro* experiments, and the EAE animal model, it is essential to acknowledge the limitations of this study. Firstly, there is a simplification in the choice of animal models: although the EAE model can simulate some pathological features of MS, its autoantigen spectrum and immune activation mechanisms differ from those in human MS, potentially affecting the clinical applicability of the study results. Secondly, the specific sites of non-degradative ubiquitination modification of PEX5 by TRIM37 and its downstream signaling pathways remain incompletely elucidated. Secondly, while our bioinformatic analysis identified multiple candidate genes (e.g., mitochondrial respiratory chain complex and macroautophagy-related genes), their functional validation and mechanistic links to the TRIM37-PEX5 axis remain unexplored. Thirdly, oxidative stress assessment was limited to ROS detection; key parameters like glutathione levels, antioxidant enzyme activities (SOD, CAT), and lipid peroxidation markers were not evaluated, leaving the antioxidant defense profile incomplete. Moreover, whether TRIM37 participates in the pathological processes of MS by regulating other peroxisomal proteins requires further validation, which may limit a comprehensive understanding of the TRIM37-PEX5 regulatory network. Additionally, the overexpression *TRIM37* relies on viral vector delivery, and its long-term safety and targeting in the CNS still need optimization. Future research should develop small molecule agonists or nanoparticle delivery systems to enhance the feasibility of clinical translation. These limitations suggest that subsequent studies should continue to explore the depth of the mechanism, the complexity of models, and the operability of therapeutic strategies to facilitate the transition of the TRIM37-PEX5 axis from basic research to clinical application.

## Conclusions

In summary, combining bioinformatics analysis and *in vitro* experiments, this study suggests the involvement of the TRIM37-PEX5 axis in MS: TRIM37 stabilizes PEX5 through non-degradative ubiquitination, potentially restoring peroxisomal function, reducing ROS-mediated apoptosis of oligodendrocytes, and promoting myelin repair.

Our research indicates: (1) low expression of TRIM37 correlates with accelerated degradation of PEX5, accompanying oxidative stress and demyelination lesions; (2) overexpression *TRIM37* improved cellular survival *in vitro* models and myelin regeneration in EAE mice; (3) the TRIM37-PEX5 axis represents a candidate target for antioxidant therapy in MS. While this study deepens mechanistic understanding, future work is needed to validate its pathophysiological and therapeutic relevance in human MS.

## Supporting information

**S1 File. Raw images underlying all gel and Western blot data presented in the manuscript.**
(PDF)

**S1 Data. Raw data underlying the quantitative results reported in the manuscript.** This file requires GraphPad Prism software (Version 10.0 or higher) for opening, to ensure data format compatibility and proper access.
(ZIP)

## Author contributions

**Data curation:** Yujia Han.

**Methodology:** Lai Jiang.

**Supervision:** Yue Jin.

**Writing – original draft:** Lai Jiang.

**Writing – review & editing:** Jin Fu.

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
