## [Decision Letter · Decision Letter 0]

27 Jul 2025

PONE-D-25-33250TRIM37-Mediated PEX5 Stabilization via Monoubiquitination Alleviates Oxidative Stress and Demyelination in Multiple SclerosisPLOS ONE

Dear Dr. Fu,

Thank you for submitting your manuscript to PLOS ONE. After careful consideration, we feel that it has merit but does not fully meet PLOS ONE’s publication criteria as it currently stands. Therefore, we invite you to submit a revised version of the manuscript that addresses the points raised during the review process.

Based on the reviewers' suggestions, the paper needs major revision. The reviewers' comments can be found below.

We look forward to receiving your revised manuscript.

Kind regards,

Tanja Grubić Kezele, Ph.D., M.D.

Academic Editor

PLOS ONE

Journal Requirements:

This research was supported by the 2024 Science and Technology Plan of the Health Commission of Heilongjiang Province (Grant No.: 20240303070234). We would like to extend our sincerest gratitude to the Health Commission of Heilongjiang Province and its relevant administrative departments. The successful execution and outcome of this project would not have been possible without your strategic support and scientific management for medical research.

The research work presented in this article was supported by the 2024 Science and Technology Plan of the Health Commission of Heilongjiang Province. The project number is 20240303070234, with Jiang Lai serving as both the project leader and the recipient of the grant. The grant was awarded by the Health Commission of Heilongjiang Province.

Reviewers' comments:

Reviewer's Responses to Questions

**Comments to the Author**

1. Is the manuscript technically sound, and do the data support the conclusions?

Reviewer #1: Yes

Reviewer #2: Yes

Reviewer #3: Partly

2. Has the statistical analysis been performed appropriately and rigorously? 

Reviewer #1: Yes

Reviewer #2: Yes

Reviewer #3: Yes

3. Have the authors made all data underlying the findings in their manuscript fully available?

Reviewer #1: Yes

Reviewer #2: Yes

Reviewer #3: Yes

4. Is the manuscript presented in an intelligible fashion and written in standard English?

Reviewer #1: Yes

Reviewer #2: Yes

Reviewer #3: Yes

5. Review Comments to the Author

Reviewer #1: Minor considerations must be addressed,

1) Title and key words must include that results are from experimental models

2) Line 41 a reference is needed after age range

3) Line 86 Caps are missing

4) Line 332 use subscript number for peroxide

Reviewer #2: Excellent reasearch article.Written according to all the rules.The study meets all ethical and integrity standards required for publication in a peer-reviewed scientific journal.The authors clearly state that all animal experiments were conducted in accordance with the ARRIVE guidelines and were approved by the Animal Ethics Committee of Beidahuang Group General Hospital (Approval No.: KY2024092003).The use of appropriate anesthesia (sodium pentobarbital) and euthanasia methods (CO₂ exposure followed by cervical dislocation) reflects compliance with internationally accepted guidelines for humane animal care and use in biomedical research.The study was funded by the Health Commission of Heilongjiang Province, and the authors explicitly declare that the funding body had no role in the study design, data analysis, publication decision, or manuscript preparation.The authors declare no competing interests, which supports the transparency and credibility of the research.All relevant data are made available within the manuscript and supplementary files, in accordance with open access data policies and good scientific practice.The methodology is sound, experiments were replicated, appropriate controls were used, and statistical analyses were correctly applied. The discussion also acknowledges limitations, which strengthens the overall reliability of the findings.

Reviewer #3: The authors present an interesting study on the regulatory role of TRIM37 in peroxidation and its impact on oxidative stress and demyelination in Multiple Sclerosis. However, substantial modifications are needed for publication:

-The context is heavily focused on the pathophysiology of MS, but no background has been provided regarding TRIM37 and peroxidation in relation to Multiple Sclerosis. Therefore, it is unclear what has previously been described and why this particular protein was chosen over others.

-The study objectives have not been outlined.

-The methodology is confusing and appears to be disorganized.

-Although the results are statistically significant, the conclusions and the implications for MS—both physiopathologically and therapeutically—are exaggerated and inconsistent with the findings. It would be advisable to moderate these conclusions.

-The limitations section does not include other relevant factors, such as those inferred from the bioinformatic analysis or the lack of assessment of additional oxidative stress parameters.

6. PLOS authors have the option to publish the peer review history of their article (what does this mean? ). If published, this will include your full peer review and any attached files.

**Do you want your identity to be public for this peer review?** For information about this choice, including consent withdrawal, please see our Privacy Policy .

Reviewer #1: **Yes: ** Alfredo Sanabria-Castro

Reviewer #2: No

Reviewer #3: No

---

## [Author Response · Author response to Decision Letter 1]

11 Sep 2025

Dear Editors,

We are submitting our research paper titled "TRIM37-Mediated PEX5 Stabilization via Monoubiquitination Alleviates Oxidative Stress and Demyelination in Multiple Sclerosis" to your esteemed journal for consideration as a "Research Article" in [PLOS One]. The authors confirm that the research content is original and has not been published or accepted for publication in any form elsewhere, nor is it being concurrently submitted to another journal. The corresponding author declares that all authors have reviewed and approved the final version of the manuscript, and that all contributors to the research have been duly acknowledged with appropriate authorship. In accordance with the journal's requirements, all original uncropped and unadjusted blot and gel images underlying the results reported in the manuscript have been included in the Supporting Information, with the file named "S1_raw_images".

Multiple sclerosis (MS) is a neurodegenerative disease characterized by chronic inflammation in the central nervous system and myelin damage. Through systematic bioinformatics analysis, our study identified a significant downregulation of the E3 ubiquitin ligase TRIM37 in MS patients, which is closely associated with disease progression. By employing an in vitro triculture model (neuron-oligodendrocyte-microglia) and in vivo experimental autoimmune encephalomyelitis (EAE) experiments, we demonstrated that overexpression of TRIM37 stabilizes the PEX5 protein through non-degradative monoubiquitination, thereby maintaining peroxisomal metabolic function, reducing oxidative stress levels, significantly decreasing apoptosis of oligodendrocytes and neurons, and promoting the expression of myelin basic protein (MBP). In vivo EAE mouse models further validated that overexpression of TRIM37 inhibits neuroinflammatory infiltration and promotes remyelination.

This study is the first to elucidate the role of TRIM37-mediated PEX5 stabilization in myelin protection, providing novel molecular insights into the pathogenesis of MS and suggesting that the TRIM37-PEX5 axis may serve as a potential therapeutic target. Given that [PLOS One] consistently publishes high-quality research that combines in-depth molecular mechanisms with translational medical value, we believe that our findings will be of great interest to your readership.

We are confident that our discovery will significantly advance the understanding of the role of ubiquitination-mediated protein homeostasis in neurodegenerative diseases. We kindly request that you consider publishing our study and look forward to receiving your valuable feedback to further improve our work.

We request to update the Funding Statement as follows:

This study was supported by the Heilongjiang Provincial Health Commission Science and Technology Program (Grant No. 20240303070234). The grant was awarded to the first author, Jiang Lai, to facilitate the study. The funders had no role in study design, data collection and analysis, decision to publish, or preparation of the manuscript. This study involves no commercial cooperation. All listed authors in this study did not receive salaries or profits from any other institutions or commercial organizations. Moreover, aside from the authors listed in the manuscript, no other sponsors or funders have played any role in this study..

Sincerely,

Jin Fu

fujin@hrbmu.edu.cn

---

## [Editor Report · Decision Letter 1]

7 Oct 2025

TRIM37-mediated stabilization of PEX5 via monoubiquitination attenuates oxidative stress and demyelination in multiple sclerosis insights from EAE and LPC-induced experimental models

PONE-D-25-33250R1

Dear Dr. Fu,

We’re pleased to inform you that your manuscript has been judged scientifically suitable for publication and will be formally accepted for publication once it meets all outstanding technical requirements.

Kind regards,

Tanja Grubić Kezele, Ph.D., M.D.

Academic Editor

PLOS ONE
---

## [Editor Report · Acceptance letter]

PONE-D-25-33250R1

PLOS ONE

Dear Dr. Fu,

I'm pleased to inform you that your manuscript has been deemed suitable for publication in PLOS ONE. Congratulations! Your manuscript is now being handed over to our production team.

Kind regards,

on behalf of

Prof. dr. Tanja Grubić Kezele

Academic Editor

PLOS ONE